# Random pruning: channel sparsity by expectation scaling factor

Chuanmeng Sun[1,2], Jiaxin Chen[1,2], Yong Li[3], Wenbo Wang[1,2] and Tiehua Ma[1,2]

[1] North University of China, State Key Laboratory of Dynamic Measurement Technology, Taiyuan, Shanxi, China
[2] North University of China, School of Electrical and Control Engineering, Taiyuan, Shanxi, China
[3] Chongqing University, State Key Laboratory of Coal Mine Disaster Dynamics and Control, Chongqing, China



## ABSTRACT

Pruning is an efficient method for deep neural network model compression and acceleration. However, existing pruning strategies, both at the filter level and at the channel level, often introduce a large amount of computation and adopt complex methods for finding sub-networks. It is found that there is a linear relationship between the sum of matrix elements of the channels in convolutional neural networks (CNNs) and the expectation scaling ratio of the image pixel distribution, which is reflects the relationship between the expectation change of the pixel distribution between the feature mapping and the input data. This implies that channels with similar expectation scaling factors ($\delta_E$) cause similar expectation changes to the input data, thus producing redundant feature mappings. Thus, this article proposes a new structured pruning method called EXP. In the proposed method, the channels with similar $\delta_E$ are randomly removed in each convolutional layer, and thus the whole network achieves random sparsity to obtain non-redundant and non-unique sub-networks. Experiments on pruning various networks show that EXP can achieve a significant reduction of FLOPs. For example, on the CIFAR-10 dataset, EXP reduces the FLOPs of the ResNet-56 model by 71.9% with a 0.23% loss in Top-1 accuracy. On ILSVRC-2012, it reduces the FLOPs of the ResNet-50 model by 60.0% with a 1.13% loss of Top-1 accuracy. Our code is available at: https://github.com/EXP-Pruning/EXP_Pruning and DOI: 10.5281/zenodo.8141065.

## INTRODUCTION

CNNs with deeper and broader structures provide higher performance for computer vision tasks. However, deeper models imply a larger number of FLOPs and parameters. For example, the original VGG-16 model (*Simonyan & Zisserman, 2014*) has hundreds of millions of parameters, and the 152-layer ResNet has gigabytes of FLOPs, which makes it difficult to deploy the models on mobile devices. To solve this problem, researchers have proposed various compression techniques for CNNs to reduce the FLOPs of the model.

The number of operations in the convolutional layer occupies 90% of the overall computation (*Yang, Chen & Sze, 2017*), so there is a large number of studies on the compression of convolutional layers. A simple approach is to construct the sparse

Corresponding author
Chuanmeng Sun,
suncm@nuc.edu.cn

convolutional layers by constraints (*Wen et al., 2016*; *Lebedev & Lempitsky, 2016*), and *He et al. (2018)* and *Li et al. (2016)* proposed a pruning method based on the norm. However, the method has a limited compression effect and does not provide significant speedup, and the weight pruning is an unstructured method that cannot be easily ported to mobile devices. Many researchers have continued to propose sophisticated solutions to the problem of exploring the importance and redundancy of filters. For example, reusing data samples to reflect the average rank (*Lin et al., 2020a*) and entropy (*Wang et al., 2021a*) of feature mappings obtained from filters to determine whether filters produce useless information; using the conditional accuracy variation associated with the results to assess the importance of each channel (*Chen et al., 2020*); calculating the classification contribution of filters to determine their importance and removing low importance filters (*Zhang et al., 2022*); using LASSO regression (*He, Zhang & Sun, 2017*) to sparse layer by layer and removing filters that are closer to the geometric median. These methods all investigate the properties that the filters have in order to explore the effect that the model internally produces on the results, however, the problem shown is that the pruning strategy is fixed leading to a large performance loss.

The above-mentioned filter property-based pruning methods do not need to add more time when developing pruning strategies. However, the adaptive pruning (*Wang, Li & Wang, 2021*; *Liu et al., 2017*; *Huang & Wang, 2018*), dynamic pruning, and architecture search-driven methods are subject to problems such as long pruning decision times. For example, using transformable architecture search to find the optimal size of small networks (*Dong & Yang, 2019*); using loss function descent to formulate weight movement rules (*Sanh, Wolf & Rush, 2020*); and combining meta-learning with architecture search (*Liu et al., 2019*). Meanwhile, *Liu et al. (2018)* pointed out that the effect of the adaptive pruning approach lies in the search for an effective network structure rather than the selection of important weights. Therefore, the stripe pruning proposed by *Meng et al. (2020)* differs from the previous structured pruning by using a filter skeleton to learn the optimal shape of the sub-network. Adaptive-based methods always add additional training conditions (using data samples or finding the best shape for a sub-network), which results in additional time costs. Therefore, the existing methods always have difficulty in finding a trade-off between high performance and simple strategies.

By analyzing the properties of the channel, this article proposes a random channel pruning method using the expectation scaling factor of the channel, and the overall pruning process is shown in Fig. 1. In the experiment, we found that for a single image sample, different channels with similar sums of matrix elements have similar altering effects on the pixel distribution of the sample, and these channels produce similar expectation ratios of pixel distribution between the feature mapping and the input data. That is, there is a linear relationship between the sum of matrix elements of the channels (unlike $\ell_1$-norm) and the $\delta_E$ of the data sample, as shown in Fig. 2. This article also assumes that the sub-networks that can effectively represent the original model capabilities are not unique, and all the parameters obtained from training play a role in the model. Therefore, this article does not select the important channels but removes the channels that produce similar effects, because randomly removing channels with similar $\delta_E$ reduces the
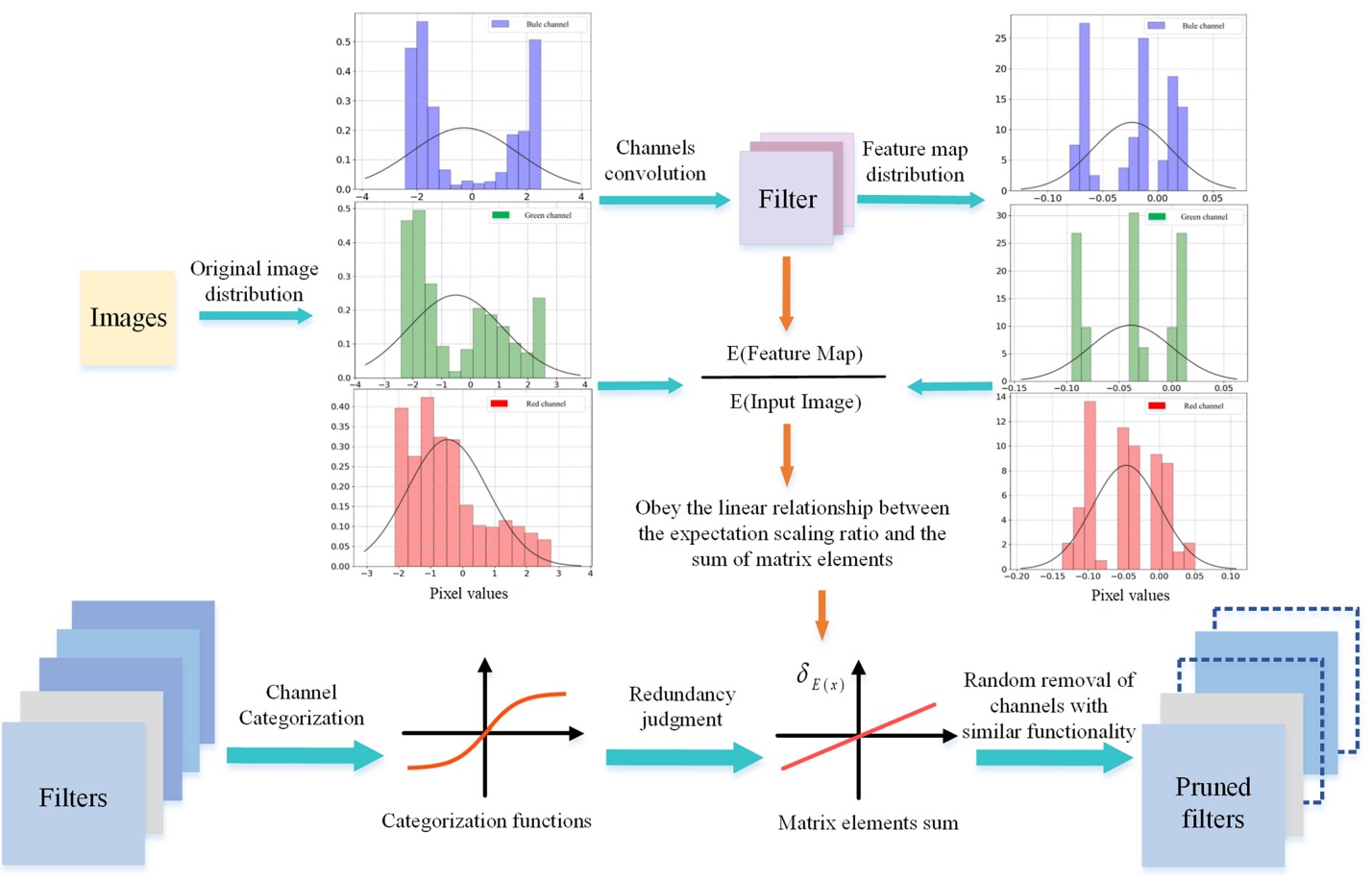

**Figure 1 Random sparsity of channels using the $\delta_E$ of the channels.** The channels are categorized using $\delta_E$ as an indicator and the linear relationship of $\delta_E$ is used to achieve stochastic sparsity for the channels.

redundancy of the model and decreases the excessive focus on local features. Moreover, the proposed EXP method is based on the model parameters, which focuses on the expectation changing effect of channels on data samples and does not introduce additional constraints, simplifying the complexity and calculation of pruning decisions. Many typical network structures, including VGGNet, ResNet, and GoogLeNet, are taken for extensive experiments on two basic datasets, namely CIFAR-10 and ImageNet ILSVRC2012. The results indicate that EXP outperforms popular pruning methods with significant compression and acceleration, and the random pruning strategy of the EXP proves that the selection of sub-networks is not unique.

In summary, the main contributions of this article are as follows.

1) Based on extensive statistical validation, it is demonstrated that for any data sample, there is always a linear relationship between the sum of matrix elements of the channels and $\delta_E$. The focus of this article on channel properties shifts from the *norm* to a change in distribution expectations.

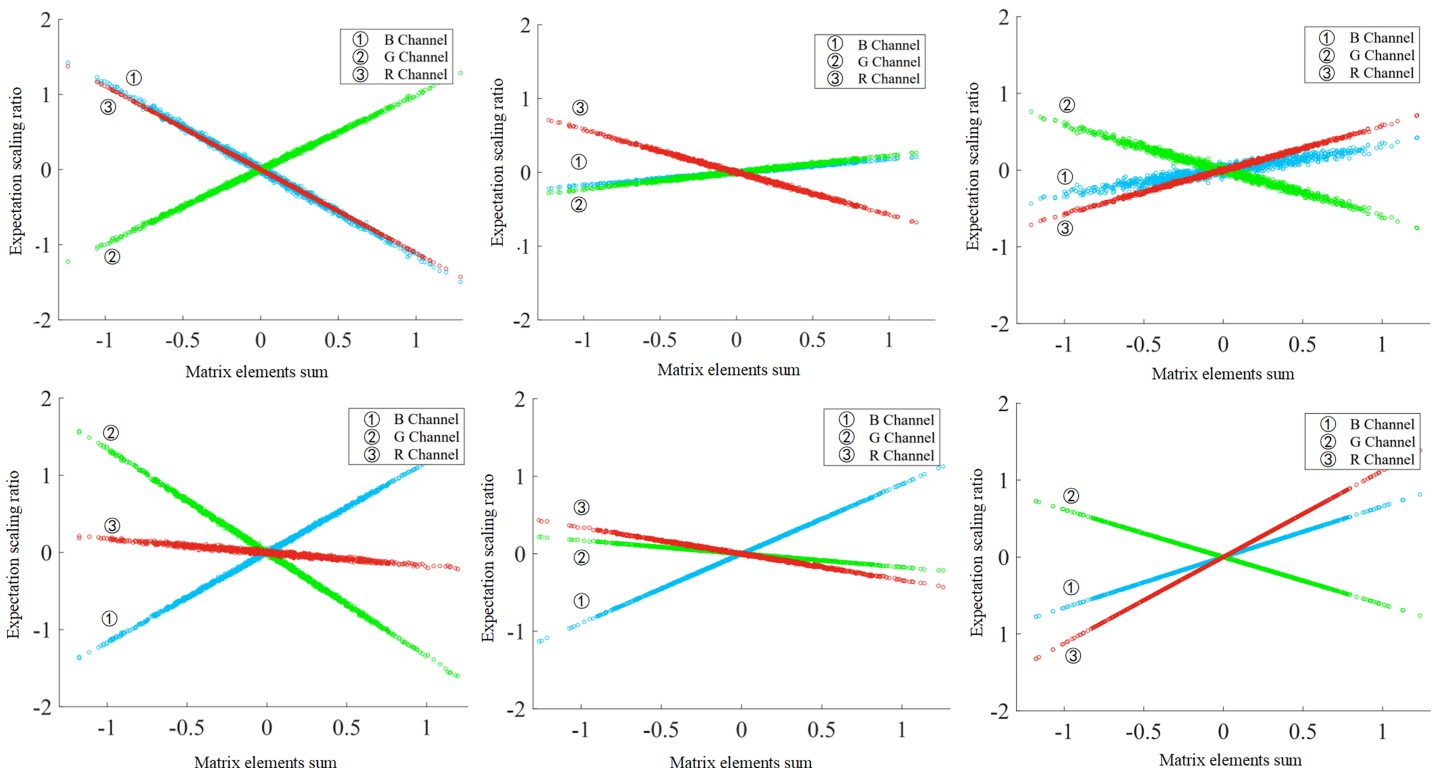

**Figure 2 Linear relationship between $\delta_E$ and $\Psi(\mathbf{w})$ of the channels.** The convolution kernel values are generated randomly. Each subgraph contains RGB channels, and $k$ in the linear relationship is affected by different input data.

2) Based on the linear relationship between the sum of matrix elements of channels and $\delta_E$, this article provides guidance for removing redundant channels to produce non-redundant and non-unique sub-networks. The advantages of the EXP in terms of compression and acceleration are demonstrated through extensive experiments and comparison with a variety of advanced methods.

## RELATED WORK

Most of the work on compressed CNNs can be divided into low-rank decomposition (*Tai et al., 2015*; *Zhang et al., 2015*), knowledge distillation (*Cheng et al., 2017*), quantization (*Son, Nah & Lee, 2018*), and pruning. Among them, pruning methods are simple and effective, and they are commonly used in model compression. To evaluate the degree of importance of filters, many empirical criteria are used to classify filters into those that contribute more and less to the network, such as based on model parameters, feature selection, parameter gradients, and architectural search-driven, as shown in Table 1.

### Based on model parameter criteria

The method relies on *a priori* knowledge to determine redundancy with the assistance of parameters in the model. For example, pruning methods based on norm information have been proposed by many researchers (*He et al., 2018*; *Li et al., 2016*; *He et al., 2019*; *He,*

**Table 1** Related work summary.

| Types | Methods | Characteristics | Articles |
|---|---|---|---|
| Model parameter criteria | Reflecting the importance of the convolution kernel based on the weight parameter | Computing is simple and relies on *a priori* knowledge | *He et al. (2018)*, *Li et al. (2016)*, *He et al. (2019)*, *Han et al. (2015)*, *Guo, Yao & Chen (2016)*, *He, Zhang & Sun (2017)*, *Yu et al. (2018)*, *Zhao et al. (2019)*, *Yang & Liu (2022)*, *Lin et al. (2021)*, *Wang et al. (2021b)* |
| Feature selection criteria | Reflecting the importance of filters based on feature mapping | Relying on partial datasets to reflect the filter's ability to extract features | *Lin et al. (2020a)*, *Tang et al. (2021)*, *Wang et al. (2021a)*, *Chen et al. (2022)*, *Yao et al. (2021)* |
| Parametric gradient criteria | Reflecting the importance of the convolution kernel based on the variation of the weight parameters during the training process | Reflecting parameter changes under the trend of model convergence, adding additional time to fine-tune the model | *Sanh, Wolf & Rush (2020)*, *Lee, Ajanthan & Torr (2018)*, *Molchanov et al. (2016)* |
| Architectural search-driven criteria | Searching for the optimal sub-network of the model from the training process | Emphasizing model structure and parameter connection, introducing intensive search process | *Dong & Yang (2019)*, *Liu et al. (2019, 2018)* |
| Others | | | *Lin et al. (2019)*, *Wang, Li & Wang (2021)*, *Tang et al. (2020)*, *Gao et al. (2021)* |

*Zhang & Sun, 2017*). *Han et al. (2015)* proposed an iterative method to remove small weights below a predefined threshold thus achieving sparsity. *Guo, Yao & Chen (2016)* proposed a dynamic pruning combined with restoration method where a partial weight is pruned and if the weight is found to be important at any time, the weight is restored. In practice, appropriate weight decay can alleviate the overfitting phenomenon, so using parameter magnitudes to determine redundancy is not reliable. In addition, the work (*Wang et al., 2021b*) proposed to use expectation and variance to directly calculate the similarity of filters. *Zhao et al. (2019)* proposed to extend the scale factor to shift terms to reformulate the batch normalization layer, estimate the channel saliency distribution and sparse it by variational inference. *Yu et al. (2018)* proposed to apply feature ranking techniques to measure the importance of each neuron in the final response layer and pruning to a binary integer optimization problem. *Lin et al. (2021)* used a message passing algorithm affinity propagation on the weight matrix to obtain an adaptive number of samples, which were then used as retained filters. *Yang & Liu (2022)* used the sum of the sensitivities of all weights in the filter to quantify the sensitivity of the filter and pruned the filter with lower 2nd order sensitivity. This type of method can simplify the pruning decision and reduce the computational complexity, and the correct prior knowledge is the key to guide the method.

### Based on feature selection criteria

This method solves the filter redundancy problem by calculating the amount of information or similarity in the feature mapping. For example, data samples are reused to reflect the average rank (*Lin et al., 2020a*) and entropy (*Wang et al., 2021a*) of the feature mappings obtained from the filters to determine whether the filters produce useless

information. Similarity between filters is analyzed using color and texture histograms of feature mappings (*Yao et al., 2021*). Using the complexity and similarity of different samples to uncover the flow pattern information of the samples, the controller is used to process the input features and predict the saliency of the channels thus completing dynamic pruning (*Tang et al., 2021*). Work (*Zhang et al., 2022*) removes filters of low importance by calculating their classification contribution. FPC (*Chen et al., 2022*) uses singular value decomposition for feature mappings to evaluate their contribution and removes the lower contributing parts. This type of method also simplifies the pruning decision, but the pruned sub-networks may be biased towards some datasets, which reduces the generalization ability.

### Based on parametric gradient criteria

The method fine-tunes the model to make changes in parameters or uses gradient information to complete pruning. Movement pruning (*Sanh, Wolf & Rush, 2020*) removes connections that gradually move away from 0 using 1st-order information of weights during fine-tuning. SNIP (*Lee, Ajanthan & Torr, 2018*) proposes a pruning method based on the importance of weight connections, which determines the importance of connections with different weights through gradient information in fine-tuning. Work (*Molchanov et al., 2016*) interleaves greedy criteria-based pruning and fine-tuning *via* backpropagation for efficient pruning. This type of method often requires constant fine-tuning of the model, which increases the time consumption of pruning decisions.

### Based on architectural search-driven criteria

The method searches for the optimal subnetwork structure of the model while training the model. These methods (*Dong & Yang, 2019*; *Liu et al., 2018*) often introduce a complex and intensive search process. PruningNet (*Liu et al., 2019*) is used to predict the parameters of the sub-networks and search for the best sub-networks using evolutionary algorithms. Artificial bee colony algorithms (*Lin et al., 2020b*) are applied to search architectures whose accuracy is considered as the fitness of each architecture.

### Others

GAL (*Lin et al., 2019*) generates sparsity by forcing the scaling factor in soft masks to zero through generative adversarial network learning. SRR-GR (*Wang, Li & Wang, 2021*) statistically models the network pruning problem and finds that pruning in the layer with the most structural redundancy outperforms pruning in the least important filter of all layers. NPPM (*Gao et al., 2021*) uses independent neural networks to predict the performance of sub-networks to maximize pruning performance as a guide and introduce situational memory to update and collect sub-networks during pruning. SCOP (*Tang et al., 2020*) prunes fake filters with larger scaling factors by training the specified network using fake and real data and mixing it with learnable scaling factors.

### Discussion

Popular pruning methods increasingly tend to be more sophisticated, but only yield smaller accuracy gains. In contrast, pruning methods based on model parameter gradient

and structural search can obtain better compression and speedup, but pay more computational time. These two problems pose fundamental challenges for deploying CNNs on mobile devices, which are attributed to the lack of theoretical guidance for determining network redundancy. In this article, we verify the effectiveness of the EXP method both theoretically and experimentally by analyzing the role of different channels in changing the expectation of the pixel distribution and exploring the redundancy rather than the importance that channels have. Compared with popular methods, the novelty of EXP is that the pruning strategy is very simple and does not introduce a computationally intensive search process, or need to consider the specificity of data samples.

## THE PROPOSED METHOD

This article aims to achieve channel-level random sparsity on the network using a linear relation for $\delta_E$. "Linear Relations" first introduces the $\delta_E$-linear relation, followed by a discussion in "Selecting Redundant Channels" on how to achieve random sparsity on the network with this linear relation.

First, notations are given to discuss the decision process of the pruning strategy. Suppose $C^i$ is the $i$-th convolutional layer in a trained CNNs and the set of all channels in this convolutional layer can be represented as $W_{C^i} = \left\{ w_1^i, w_2^i, \ldots, w_r^i, \ldots, w_{R_i}^i \right\}$ $\in \mathbb{R}^{R_i \times l_i \times l_i}$. In the pruning process, the channels in each convolutional layer are divided into two groups, namely the subset $A = \left\{ w_A^1, w_A^2, \ldots, w_A^S \right\}$ to be retained and the subset $B = \left\{ w_B^1, w_B^2, \ldots, w_B^T \right\}$ to be deleted. $S$ and $T$ are respectively the number of channels to be retained and to be deleted, $A \bigcap B = \varnothing$, $A \bigcup B = W_{C^i}$, $S + T = R_i$.

### Linear relations

For any input image $X \in \mathbb{R}^{M \times N}$ and any convolution kernel (channel) $w \in \mathbb{R}^{U \times V}$, then the feature mapping $Y \in R^{M \times N}$ can be obtained from $Y = w \otimes X$ ($\otimes$ denotes the convolution operation). If $X$ and $Y$ are regarded as random variables, then the expectation of $X$ and $Y$ are $\mu$ and $\mu'$, respectively. In fact, the convolution operation changes the expectation of the image as follows

$$\mu' = E(\mathbf{Y}) = E(\mathbf{w} \otimes \mathbf{X}) = k \cdot \Psi(\mathbf{w}) \cdot E(\mathbf{X}) = k \cdot \Psi(\mathbf{w}) \cdot \mu \tag{1}$$

$$\Psi(\mathbf{w}) = k \cdot \frac{\mu'}{\mu} = k \cdot \delta_E \tag{2}$$

where $\Psi(\cdot)$ denotes the summation of elements and there is a linear relationship between $\delta_E$ and $\Psi(\mathbf{w})$. In visual inspection, different convolution kernels implement a variety of operations on the image. Convolutional operations achieve a scaling of the distributional expectation. The distribution expectation reflects the overall information of the feature mapping, so the expectation scaling factor $\delta_E$ works as a basic feature for the channel to reflect the ability to extract features.

In this article, a randomized convolutional kernel experiment is used to validate the relationship between $\delta_E$ of data samples and $\Psi(\mathbf{w})$ of channels. The randomly selected convolutional kernels have different $\Psi(\mathbf{w})$, and the normalized images from the CIFAR-10 dataset are used as input. As illustrated in Fig. 2, the $\Psi(\mathbf{w})$ of the channel can directly affect

the change in the expectation value of the distribution, *i.e.*, the value of $\delta_E$, for any data sample. This indicates that $\delta_E$ and $\Psi(\mathbf{w})$ follow an approximately linear relationship. This direct linear relationship provides a theoretical guide to the selection of redundant channels.

## Selecting redundant channels

Any channel with similar $\Psi(\mathbf{w})$ can generate similar $\delta_E$ in different data samples. This article argues that similar $\delta_E$ is the cause of network redundancy, so the channels need to be sparse to reach the role of reducing similar underlying features. And the pruning process should make $\delta_E$ balanced across scales to maintain the rich feature extraction capability of the network.

The pruning strategy in this article aims to identify and sparse the redundant set of channels from $W_{C^i}$. In order to make the features extracted from the original network be effectively retained at different scales, first all channels are divided into several scales and the channels are randomly pruned at the same pruning rate at each scale. Uniformly sparse the $\delta_E$ of convolutional layers can make the sum of channel weights $\Psi(W_{C^i})$ of the original channel set $W_{C^i}$ have the same distribution as the sum of channel weights $\Psi(A)$ of the retained channel set $A$, so that the accuracy and generalization performance of the pruned model can be maintained. This article will implement pruning on the model from two aspects, *i.e.*, considering the effects of global pruning and local pruning.

**Local pruning.** For local pruning (called EXP-A), different filters have divided the set $W_{C^i}$. Let the number of filters be $M$, then

$$W_{C^i} = P_1 \cup \cdots \cup P_m \cup \cdots \cup P_M \qquad (3)$$

where, $P_m$ denotes the $m$-th filter, $1 \le m \le M$. Assuming that the class of basic features (non-redundant features) present in the filter $P_m$ is $N_m^{BF}$, then pruning is to remove the redundant channels of each basic feature in $P_m$. In the model, a similarity evaluation function $S_{\text{index}}(\cdot)$ is required for the judgment of similarity $\Psi(\mathbf{w})$, so as to effectively characterize the redundancy of a certain set of channels $D$ as follows.

$$S_{\text{index}}\left(d^i\right) = \text{round}\left[f\left(d^i\right)\right] = \text{round}\left[N \cdot \left(\frac{1}{1 + \exp\left(-\frac{\Psi(d^i)}{\max(\Psi(\mathbf{D}))}\right)} - \frac{1}{2}\right)\right] \qquad (4)$$

where $d^i$ is the $i$-th channel in the channel set $D$; round$(\cdot)$ represents rounding calculation; $N$ plays the role of regulating the granularity of similarity, the larger $N$ is, the more similar classes ( basic feature types $N^{BF}$) are divided in the set $D$, the stricter the conditions for judging similarity. Equation (4) essentially divides channels into N + 1 classes, and one class represents one basic feature, so the number of basic feature types contained in $D$ is $N_D^{BF} = N + 1$.

Using the similarity evaluation function $S_{\text{index}}(\cdot)$, we can count the number of redundant channels contained in each class of the basic features in $P_m$. Let $P_m^{n_{BF}}$ denote the set of channels of the $n_{BF}$-th class of basic features in $P_m$, $1 \le n_{BF} \le N_m^{BF}$; $w_{m,n_{BF}}^{r'}$ denote the $r'$-th channel in $P_m^{n_{BF}}$, $1 \le r' \le R'$. Then the redundancy of $P_m^{n_{BF}}$ is defined as:

$$\mathrm{Re}\left(P_m^{n_{BF}}\right) = \sum_{r'=1}^{R'} \mathrm{assign}\left(w_{m,n_{BF}}^{r'}\right) \tag{5}$$

$$\mathrm{assign}\left(w_{m,n_{IF}}^{r'}\right) = \begin{cases} 1, S_{\mathrm{index}}\left(w_{m,n_{IF}}^{r'}\right) \in \left\{ S_{\mathrm{index}}\left(w_{m,n_{IF}}^{r'+1}\right), \cdots, S_{\mathrm{index}}\left(w_{m,n_{IF}}^{R'}\right) \right\} \\ 0, S_{\mathrm{index}}\left(w_{m,n_{FF}}^{r'}\right) \notin \left\{ S_{\mathrm{index}}\left(w_{m,n_{IF}}^{r'+1}\right), \cdots, S_{\mathrm{index}}\left(w_{m,n_{IF}}^{R'}\right) \right\} \end{cases} \tag{6}$$

Therefore, the redundant channels can be randomly pruned according to the set compression rate $\alpha$ ($0 \le \alpha \le 1$) for $P_m^{n_{BF}}$. The percentage of deleted channels is:

$$\rho_{del}\left(P_m^{n_{BF}}\right) = \alpha \cdot \frac{\mathrm{Re}\left(P_m^{n_{BF}}\right)}{R'} \tag{7}$$

The final pruning process in the EXP-A method is as follows: (i) Set the channel granularity factor $N$ and pruning rate $\alpha$; (ii) Categorize all channels in $P_m$ into N + 1 basic feature subsets one by one by Eq. (4), and sort them by the number of elements in the subsets. The subset with the highest number is $P_{m,1}^{n_{BF}}$, and the one with the lowest number is $P_{m,N+1}^{n_{BF}}$. (iii) Calculate the redundancy $Re\left(P_{m,j}^{n_{BF}}\right)$ and the deletion channel ratio $\rho_{del}\left(P_{m,j}^{n_{BF}}\right)$ of each subset in turn based on Eqs. (5) and (7), and delete each subset channel randomly according to $\rho_{del}\left(P_{m,j}^{n_{BF}}\right)$. The deletion process is as follows: each channel in $P_{m,j}^{n_{BF}}$ is examined in turn, and a random number *rand* with the interval in [0,1] is generated at each examination, and if $rand \le \rho_{del}\left(P_{m,j}^{n_{BF}}\right)$, the channel is deleted; otherwise, it is retained.

This article notes that the direction of abstraction of CNNs from the underlying features to higher-order features is directed to the same semantic features. This means that the number of types of basic features under each scale in each convolutional layer should be the same, *i.e.*, for the set of channels $W_{C^i}$, which contains the same number of basic features as its subset $P_m$ of channels under each scale, there is:

$$N^{BF} = N_1^{BF} = \ldots = N_m^{BF} = \ldots = N_M^{BF} = N + 1 \tag{8}$$

**Global pruning.** Global pruning (called EXP-B) performs random pruning of all channels $W_{C^i}$ in the convolutional layer. From "Non-uniqueness and Stability of Sub-networks", it can be seen that $\Psi(\mathbf{w})$ in each convolutional layer obeys Gaussian distribution. In order to make the basic features in the channel set are effectively retained in different scales, $\Psi(\mathbf{w})$ is uniformly distributed with the help of Eq. (4). Meanwhile, the similarity evaluation function maps the channel weights and $\Psi(\mathbf{w})$ to the integers in the $[-N/2, N/2]$ interval, *i.e.*, $S_{index}(\cdot)$ has the effect of uniformly classifying the $\Psi(\mathbf{w})$ scales. It can be seen that the pruning using the similarity evaluation function $S_{index}(\cdot)$ contains the grading of scales, in which the scales are classified into N + 1 levels, *i.e.*, $\Psi(\mathbf{W_{C^i}})$ can be regarded as consisting of N + 1 sets. Thus the EXP-B pruning method replaces the filter $P_m$ with the set of channels $W_{C^i}$ only in step 2 compared to EXP-A.

After random sparse, the set of channels with similar $\delta_E$ retains only some of the basic features, thus achieving random channel pruning of CNNs layer by layer. This differs from pruning methods using data samples or adaptive pruning, and it helps to save much time

in the selection of channels. Also random pruning means that whatever changes the channels make to the distribution expectations of the data samples, this article completes pruning by removing only the channels with similar $\delta_E$.

# EXPERIMENT

## Experimental settings

This article validates the proposed method using CIFAR-10 (*Torralba, Fergus & Freeman, 2008*) and ImageNet ILSVRC2012 (*Russakovsky et al., 2015*) datasets to investigate the efficiency of this method with other methods in reducing model complexity. And it is also tested for networks with different structures, including VGGNet (*Simonyan & Zisserman, 2014*), ResNet (*He et al., 2016*), and GoogLeNet (*Szegedy et al., 2015*). The complexity and performance of the models were evaluated using floating-point operations and TOP-1 accuracy. All experiments were trained and tested on an NVIDIA Telsa P40 graphics card using the Pytorch (*Paszke et al., 2017*) architecture.

The stochastic gradient descent algorithm (SGD) was adopted to solve the optimization problem. The training batch size was set to 128, the weight decay was 0.0005, and the momentum was set to 0.9. On the CIFAR-10 dataset, fine-tuning was performed with 30 epochs; the initial learning rate was 0.01 and decayed by dividing by 10 at epochs 15 and 30. On the ImageNet dataset, fine-tuning was performed with 20 epochs; the initial learning rate was 0.001 and decayed by dividing by 10 at epochs of 15 and 25.

## Results and analysis

### Results of CIFAR-10

In the dataset, the image size is $32 \times 32$, and there are 10 categories with a total of 50,000 training images and 10,000 test images. The method proposed in this article is compared with the mainstream pruning methods, including VB (*Zhao et al., 2019*), GAL (*Lin et al., 2019*), HRank (*Lin et al., 2020a*), NISP (*Yu et al., 2018*), GM (*He et al., 2019*), TAS (*Dong & Yang, 2019*), SRR-GR (*Wang, Li & Wang, 2021*), ManiDP (*Tang et al., 2021*), NPPM (*Gao et al., 2021*), SCOP (*Tang et al., 2020*), CAC (*Chen et al., 2020*), SENS (*Yang & Liu, 2022*), FPC (*Chen et al., 2022*), and EPruner (*Lin et al., 2021*). Tables 2–4 show the results of multiple methods on CIFAR-10. According to the experience, setting the value of *N* in Eq. (4) to 13 on this dataset enables the redundant categories to be clearly delineated and contributes to a higher pruning rate.

**VGG-16.** The EXP method maintains a high accuracy despite the large reduction in FLOPs. EXP-B achieves a 60.8% reduction in FLOPs with a 0.23% reduction in accuracy compared to the baseline model. In contrast, the SENS (*Yang & Liu, 2022*) based on the model parameter criteria, which uses 2nd-order sensitivity to remove insensitive filters, only results in a 54.1% reduction in FLOPs while decreasing accuracy by 0.53%. When achieving a 70.89% compression of FLOPs, the EXP-B leads to an accuracy reduction of only 0.47%.

**ResNet-56/110.** In ResNet-56, EXP-B reduces the FLOPs by 60.5%, and the accuracy increases by 0.37% compared to the baseline model. When larger compression is achieved, the FLOPs decrease by 71.9%, and the accuracy decreases by only 0.23%. Compared to the

**Table 2  Pruning results of VGGNet-16 on CIFAR-10.**

| Method | Baseline | Top-1 Acc. | Acc. ↑↓ | FLOPs ↓ |
|---|---|---|---|---|
| VB (2019) | 93.25% | 93.18% | −0.07% | 39.1% |
| GAL-0.05 (2019) | 93.96% | 92.03% | −1.93% | 39.6% |
| EXP (ours)-A | 93.96% | 93.64% | −0.32% | 53.1% |
| EXP (ours)-B | 93.96% | 93.55% | −0.41% | 53.1% |
| HRank (2020) | 93.96% | 93.43% | −0.53% | 53.5% |
| SENS (2022) | 93.70% | 93.17% | −0.53% | 54.1% |
| EXP (ours)-A | 93.96% | 93.50% | −0.46% | 60.8% |
| EXP (ours)-B | 93.96% | 93.73% | −0.23% | 60.8% |
| HRank (2020) | 93.96% | 92.34% | −1.62% | 65.3% |
| EXP (ours)-A | 93.96% | 92.50% | −1.46% | 70.9% |
| EXP (ours)-B | 93.96% | 93.54% | −0.42% | 70.9% |

feature selection-based HRank (*Lin et al., 2020a*), the EXP can effectively compress the model while maintaining stable performance. SRR-GR (*Wang, Li & Wang, 2021*) removes the most redundant filters from the network by calculating the filter redundancy score, however, the method uses L2 norm as the criterion for filter redundancy, and therefore the pruning results have mediocre performance. In ResNet-110, EXP-B also showed better compression performance than FPC (*Chen et al., 2022*), with a 70.0% reduction in FLOPs and a 0.28% improvement in accuracy over the baseline model.

**GoogLeNet.** The results show that EXP-A can reduce the FLOPs by 70.4% with only a 0.03% decrease in accuracy. EPruner (*Lin et al., 2021*) used the transfer algorithm Affinity Propagation for calculating an adaptive number of samples, which then act as a preserved filter. The EXP method achieves comparable performance to this method, but the pruning method is simpler and easier to implement.

### Results of ImageNet

The ImageNet ILSVRC2012 classification dataset with 1,000 classes is a more challenging dataset. It contains 1.28 million training images and 50,000 test images, with an image size of 224 × 224. The comparison of the proposed method with other popular model pruning methods, including GAL (*Lin et al., 2019*), HRank (*Lin et al., 2020a*), GM (*He et al., 2019*), TAS (*Dong & Yang, 2019*), SRR-GR (*Wang, Li & Wang, 2021*), SCOP (*Tang et al., 2020*), EPruner (*Lin et al., 2021*), SENS (*Yang & Liu, 2022*), WB (*Zhang et al., 2022*), and L1-*norm* (*Wang et al., 2023*), also shows the superiority of our method, as shown in Table 5.

ResNet-50 has more parameters than ResNet-56, and to clearly distinguish the class to which the parameters belong, the value of $N$ in Eq. (4) is set to 15. EXP-B resulted in a TOP-1 accuracy of 75.76% when FLOPs were reduced by 53.1%. The SCOP-B (*Tang et al., 2020*) based on scientific control achieved a similar compression of FLOPs while the accuracy was reduced by 0.89%. The WB (*Zhang et al., 2022*) based on feature selection preserves the channels that contribute to most categories by visualizing feature mapping, and this method achieves 63.5% FLOPs reduction with 1.94% accuracy reduction. In

**Table 3  Pruning results of ResNet-56 and ResNet-110 on CIFAR-10.**

|  | Method | Baseline | Top-1 Acc. | Acc. ↑↓ | FLOPs ↓ |
|---|---|---|---|---|---|
| ResNet-56 | FPC (2022) | 93.78% | 93.39% | −0.39% | 49.7% |
|  | GM (2019) | 93.59% | 92.93% | −0.66% | 52.6% |
|  | NPPM (2021) | 93.04% | 93.40% | +0.36% | 50.0% |
|  | EXP (ours)-A | 93.26% | 93.54% | +0.28% | 53.5% |
|  | EXP (ours)-B | 93.26% | 93.82% | +0.56% | 53.5% |
|  | SCOP (2020) | 93.70% | 93.64% | −0.06% | 56.0% |
|  | TAS (2019) | 94.46% | 93.69% | −0.77% | 52.7% |
|  | SRR-GR (2021) | 93.38% | 93.75% | +0.37% | 53.8% |
|  | SENS (2022) | 93.78% | 93.17% | −0.53% | 54.1% |
|  | ManiDP (2021) | N/A | 93.64% | N/A | 62.4% |
|  | EXP (ours)-A | 93.26% | 93.48% | +0.22% | 60.5% |
|  | EXP (ours)-B | 93.26% | 93.63% | +0.37% | 60.5% |
|  | EPruner (2021) | 93.26% | 93.18% | −0.08% | 61.3% |
|  | HRank (2020) | 93.26% | 90.72% | −2.54% | 74.1% |
|  | EXP (ours)-A | 93.26% | 92.91% | −0.35% | 71.9% |
|  | EXP (ours)-B | 93.26% | 93.03% | −0.23% | 71.9% |
| ResNet-110 | GAL-0.5 (2019) | 93.50% | 92.55% | −0.95% | 48.5% |
|  | CAC (2021) | 93.38% | 93.86% | +0.48% | 50.0% |
|  | EXP (ours)-A | 93.50% | 93.96% | +0.46% | 60.1% |
|  | EXP (ours)-B | 93.50% | 93.73% | +0.23% | 60.1% |
|  | FPC (2022) | 94.43% | 93.83% | −0.60% | 64.5% |
|  | EXP (ours)-A | 93.50% | 93.52% | +0.02% | 70.0% |
|  | EXP (ours)-B | 93.50% | 93.78% | +0.28% | 70.0% |

**Table 4  Pruning results of GoogLeNet on CIFAR-10.**

| Method | Baseline | Top-1 Acc. | Acc. ↑↓ | FLOPs ↓ |
|---|---|---|---|---|
| GAL-0.05 (2019) | 95.02% | 93.93% | −1.12% | 38.2% |
| HRank (2020) | 95.05% | 94.53% | −0.52% | 54.9% |
| EXP (ours)-A | 95.05% | 95.02% | −0.03% | 62.1% |
| EXP (ours)-B | 95.05% | 94.95% | −0.10% | 62.1% |
| EPruner (2021) | 95.05% | 94.99% | −0.06% | 67.3% |
| EXP (ours)-A | 95.05% | 95.02% | −0.03% | 70.4% |
| EXP (ours)-B | 95.05% | 94.87% | −0.18% | 70.4% |

contrast, the EXP-B method only resulted in a 1.13% decrease in TOP-1 accuracy when achieving a 60.6% decrease in FLOPs, from which it can be concluded that the method based on model parameters can be adapted to different datasets.

**Table 5  Pruning results of ResNet-50 on ImageNet ILSVRC2012.**

| Method | Baseline | TOP-1 Acc. | Acc. ↓ | TOP-5 Acc. | FLOPs ↓ |
|---|---|---|---|---|---|
| GM (2019) | 76.15% | 75.59% | 0.56% | 92.63% | 42.2% |
| TAS (2019) | 76.20% | 74.94% | 1.26% | 92.59% | 43.5% |
| EXP (ours)-A | 76.15% | 75.71% | 0.44% | 92.60% | 53.1% |
| EXP (ours)-B | 76.15% | 75.76% | 0.39% | 92.50% | 53.1% |
| GAL-1 (2019) | 76.15% | 69.88% | 6.27% | 89.75% | 61.3% |
| SCOP-B (2020) | 76.15% | 75.26% | 0.89% | 92.53% | 54.6% |
| HRank (2020) | 76.15% | 71.98% | 4.17% | 91.01% | 62.1% |
| SRR-GR (2021) | 76.13% | 75.11% | 1.02% | 92.35% | 55.1% |
| EPruner (2021) | 76.01% | 74.26% | 1.75% | 92.96% | 53.3% |
| SENS (2022) | 76.15% | 75.23% | 0.92% | N/A | 56.3% |
| WB (2022) | 76.15% | 74.21% | 1.94% | 92.01% | 63.5% |
| EXP (ours)-A | 76.15% | 74.53% | 1.62% | 92.07% | 60.6% |
| EXP (ours)-B | 76.15% | 75.02% | 1.13% | 92.40% | 60.0% |
| L1-*norm* (2023) | 76.15% | 74.77% | 1.38% | N/A | 60.9% |

## Discussion

EXP is based on model parameters and aims to investigate the effect of randomly removing channels with similar $\delta_E$ on the results. Compared to other pruning methods based on model parameters (*Yu et al., 2018*; *Zhao et al., 2019*; *Yang & Liu, 2022*; *Lin et al., 2021*; *Wang et al., 2023*), the EXP is simpler and more effective by calculating the sum of weights of convolutional kernels and using a similarity evaluation function for redundancy class determination, and finally randomly sparse the set of redundant features to obtain non-unique sub-networks. In traditional pruning work, the *norm* is commonly used to analyze the importance of the convolutional kernels, while in the work (*Wang et al., 2023*) the learning schedule and learning rate decay rules are analyzed to reconceptualize the effectiveness of the L1-*norm* for filter pruning. In the ImageNet ILSVRC2012 dataset, the EXP-A method slightly underperforms the results of L1-*norm*, while EXP-B improves the Top-1 accuracy by 0.25% in comparison, and this article provides a new idea for analyzing the redundancy properties of convolutional kernels.

The feature selection-based methods (*Zhang et al., 2022*; *Chen et al., 2022*) show a larger loss due to the fact that the selected sub-networks are more biased towards part of the dataset and the sub-dataset when performing feature selection has a distribution bias from the original dataset. NPPM (*Gao et al., 2021*) and SCOP (*Tang et al., 2020*), on the other hand, introduce additional operations to determine the pruning decision, incurring more computational resource consumption. The EXP method, however, is independent of the dataset and is simpler to formulate pruning decisions.

Comparing the results of local pruning (EXP-A) with global pruning (EXP-B) from various aspects, EXP-B shows a better balance. In terms of parameter connectivity, EXP-B has more opportunities to retain the better model connectivity, which is more flexible compared to EXP-A, where one is not restricted to have to connect to a certain filter. In

terms of computation, while EXP-A computes the redundancy in the filters one by one, the EXP-B method computes the global redundancy directly, saving more computation time. In terms of the distribution of the sum of convolutional kernel weights, the EXP-A method has a smaller range for redundancy discrimination, while the distribution of the sub-networks obtained by EXP-B, which involves global discrimination, is closer to the original distribution.

## Non-uniqueness and stability of sub-networks
### Non-uniqueness

The proposed EXP method generates different sub-networks by randomly pruning the set of redundant channels. Figure 3 shows the distribution of the channel matrix element sums for the 16-th convolutional layer in the pre-trained ResNet-56 model and the distribution after random pruning. The original distribution approximately obeys a Gaussian distribution with skewness, and after various degrees of compression, the randomly preserved subnetworks still approximately obey a Gaussian distribution. One of the reasons why the performance of the subnetwork remains stable is that the convolutional layer distribution of the sparse subnetwork is the same as the original distribution. Pruning using the norm attribute leads to a dispersion of the parameters into two clusters (away from 0), making the parameter distribution discontinuous. In contrast, sparse redundant channels result in a smoother distribution of parameters, which still cover the entire interval but only achieve different degrees of density reduction.

### Stability

The experimental results in "Results and Analysis" show that random sparsity does not reduce the performance of the retained subnetwork. Another reason why the subnetwork performance remains stable is that the redundant channels overly focus on the features of the data samples, and the removal of redundant channels introduces generalization effects by weakening the focus on features rather than ignoring them. Table 6 shows the results of multiple random repetition experiments are shown, and the results indicate that random removal of redundant channels can also keep the pruning results stable. For example, ResNet-56 maintains accuracy of about 93.60% after a 60.55% reduction in FLOPs, producing fluctuations that can be controlled within 0.10%.

## Generalization impact

It is considered that channels with similar $\delta_E$ will overly focus on local features, thus causing an overfitting phenomenon. Meanwhile, making the redundant channels sparse can improve the generalization ability of the model. Taking the ResNet-56 model as an example, experiments were conducted using EXP-A and EXP-B at different compression rates, and the results are shown in Fig. 4A. It can be seen that the accuracy is always maintained above the baseline level at the pruning rates of 0.53 and 0.60, and the continuous removal of redundant channels can continuously improve the generalization ability of the model to enable the accuracy to be continuously improved. Works (*Bartoldson, Barbu & Erlebacher, 2018*; *Bartoldson et al., 2020*) have shown that pruning on the later layers is sufficient to improve the generalization ability of the model. To

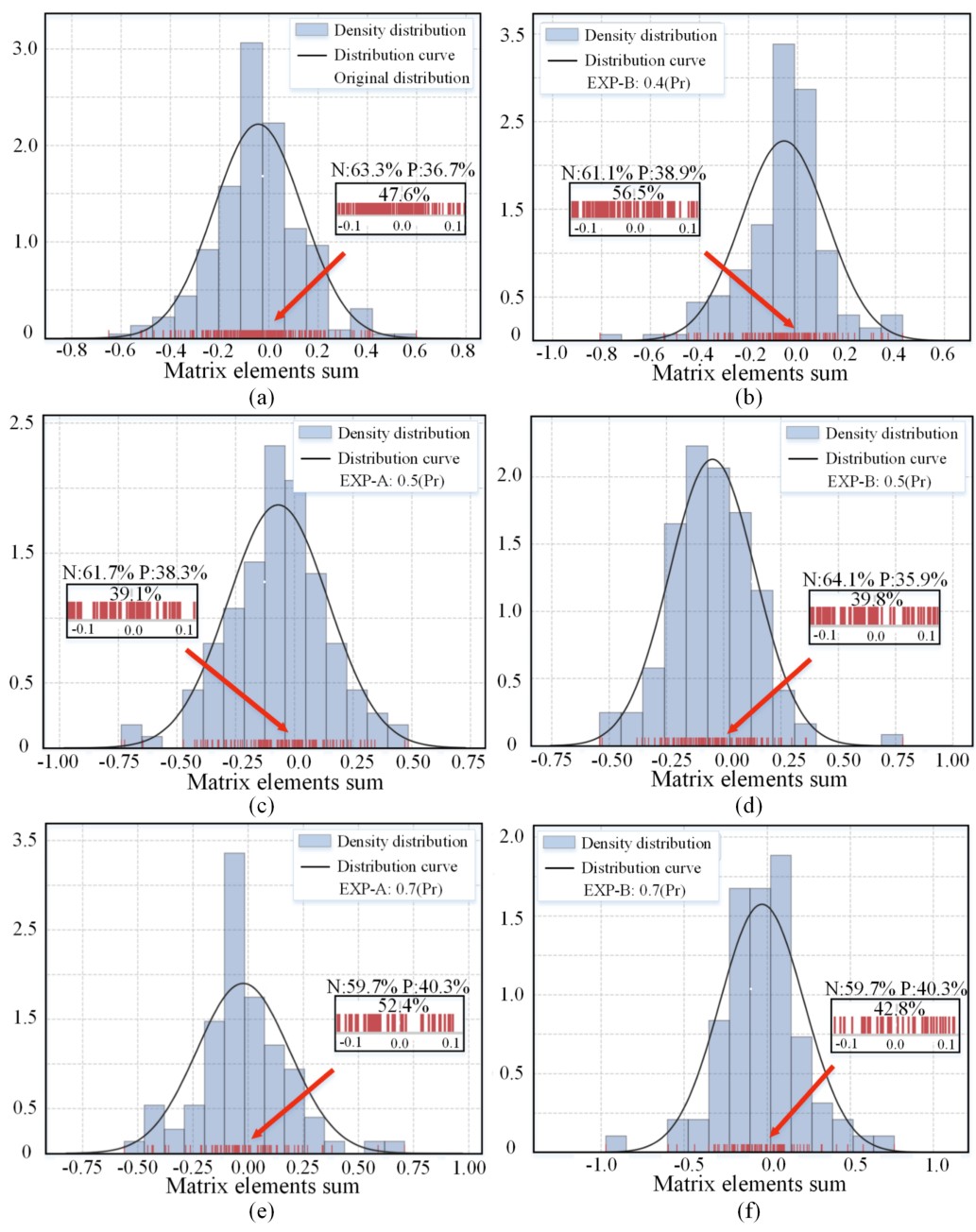

**Figure 3 (A–F) Distribution of the sum of matrix elements $\Psi(W_{C^i})$ of the channels in the 16-th convolutional layer of the pre-trained and pruned ResNet-56 networks.** The red lines indicate the locations of the specific values of the sum of elements. The three percentage values correspond to the percentage of values between −0.1 and 0.1, positive percentage (P), and negative percentage (N). Pr denotes the pruning rate.

achieve high compression rates, less compression of the earlier layers and excessive compression of the later layers will cause a decrease in the generalization ability of the model. Figure 4B shows that setting a larger pruning rate for the later layers of the model will destroy the high-level features of the model and make the network performance degrade rapidly. Therefore, the generalization ability cannot be effectively improved, and

**Table 6 Results of ResNet-56 multiple replication experiments on CIFAR-10.**

| Method | TOP-1 Acc. | TOP-1 Acc. ↑↓ | TOP-5 Acc. | FLOPs ↓ |
| --- | --- | --- | --- | --- |
| ResNet-56 | 93.26% | N/A | 99.78% | N/A |
| EXP-A | 93.48 ± 0.12% | +0.22 ± 0.12% | 99.76 ± 0.15% | 60.55% |
| EXP-B | 93.63 ± 0.10% | +0.37 ± 0.10% | 99.77 ± 0.15% | 60.55% |
| EXP-A | 92.91 ± 0.12% | −0.35 ± 0.12% | 99.81 ± 0.15% | 71.91% |
| EXP-B | 93.00 ± 0.10% | −0.23 ± 0.10% | 99.78 ± 0.15% | 71.91% |

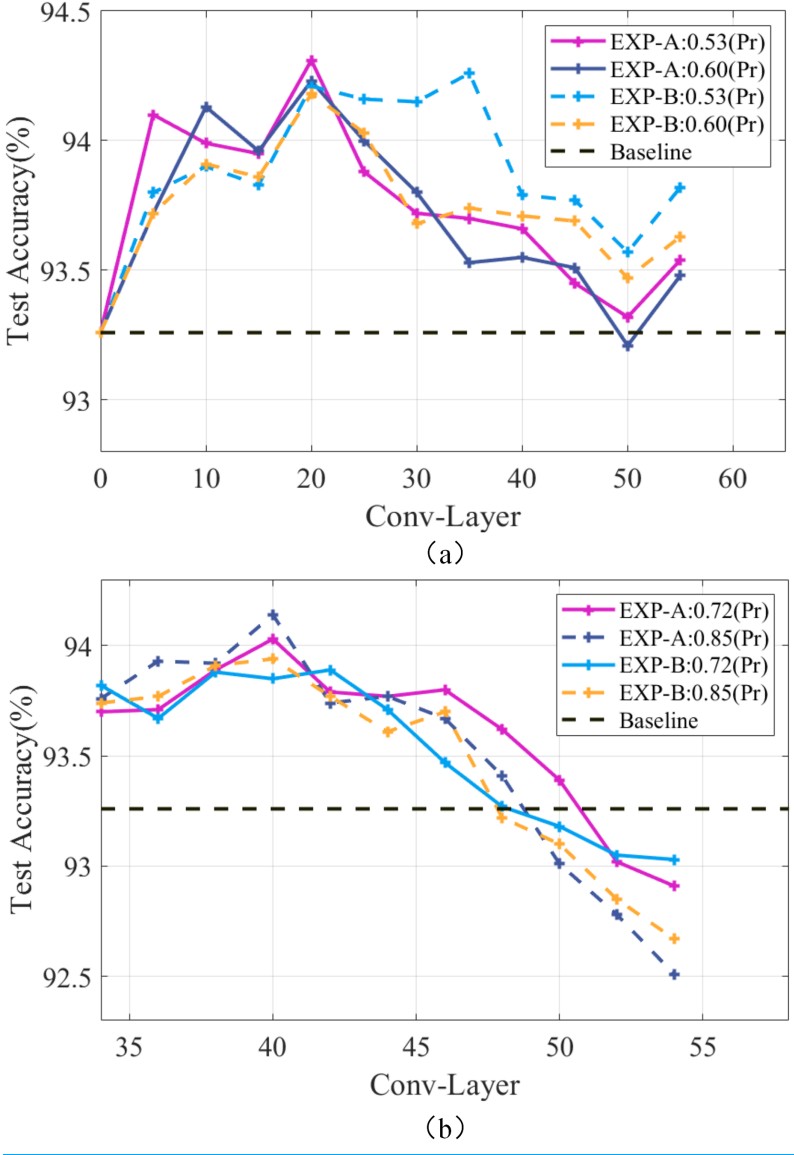

**Figure 4 Performance of the ResNet-56 network at different compression rates.** Pr denotes the pruning rate and Conv-Layer denotes the convolutional layer.

the accuracy of the model tends to decrease. So, the pruning rate should be set as globally smooth as possible.

## CONCLUSION

Based on the discovery of a linear relationship between the sum of matrix elements of channels and the expectation scaling factor of the pixel distribution, this article proposes a new structured pruning method called EXP. This method uses the linear relationship to establish channels with similar $\delta_E$ as a redundant channel set. By randomly pruning this set of channels, the excessive focus on local features by redundant channels can be weakened, thus obtaining non-redundant and non-unique sub-networks. Through extensive experiments and analysis, the effectiveness of the proposed EXP method is verified. Future work will focus on the generalization effects caused by sparsity to optimize DNNs.

### Funding

This work was supported by the National Key Research and Development Program of China (2022YFC2905700), the National Key Research and Development Program of China (2022YFB3205800), and the Fundamental Research Programs of Shanxi Province (202103021224199, 202203021221106). The funders had no role in study design, data collection and analysis, decision to publish, or preparation of the manuscript.

### Grant Disclosures

The following grant information was disclosed by the authors:
National Key Research and Development Program of China: 2022YFC2905700.
National Key Research and Development Program of China: 2022YFB3205800.
Fundamental Research Programs of Shanxi Province: 202103021224199, 202203021221106.

### Competing Interests

The authors declare that they have no competing interests.

### Author Contributions

- Chuanmeng Sun conceived and designed the experiments, analyzed the data, authored or reviewed drafts of the article, and approved the final draft.
- Jiaxin Chen conceived and designed the experiments, performed the experiments, performed the computation work, prepared figures and/or tables, authored or reviewed drafts of the article, and approved the final draft.
- Yong Li conceived and designed the experiments, analyzed the data, performed the computation work, authored or reviewed drafts of the article, and approved the final draft.
- Wenbo Wang conceived and designed the experiments, performed the experiments, performed the computation work, prepared figures and/or tables, and approved the final draft.

- Tiehua Ma conceived and designed the experiments, analyzed the data, authored or reviewed drafts of the article, and approved the final draft.

## Data Availability

The code is available at GitHub and Zenodo:

- https://github.com/EXP-Pruning/EXP_Pruning.

- EXP. (2023). EXP-Pruning/EXP_Pruning: v1.0.0 (python). Zenodo. https://doi.org/10.5281/zenodo.8141065.

The data is available at The CIFAR-10 dataset.

(http://www.cs.toronto.edu/~kriz/cifar.html) and ImageNet Large Scale Visual Recognition Challenge 2012 (ILSVRC2012).

(https://www.image-net.org/challenges/LSVRC/2012/index.php).

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
