# Peer review of "Random pruning: channel sparsity by expectation scaling factor"

_PeerJ Computer Science, doi:10.7717/peerj-cs.1564_

## Round 0.1 · original submission · Minor Revisions

Please review the comments made by the reviewers, make appropriate and necessary changes, and submit the revised version and an explanation of how each of the comments was taken care of. I will review the changes made and your explanation and decide if the paper is ready to be published. Please make sure that you also have gone through the writing style and address all language-related issues.

Reviewer 1 ·

Basic reporting

First of all, the reviewer enjoyed reading this article (the reviewer has experience working on pruning) and appreciated the extensive experiments to support the proposed claims. The writing is clear at most places and the summary of several methods in the “Related Work” section is adequate enough for new readers to understand different pruning methods. The figures help quickly grasp the methodologies in the paper. Probably, the authors can add a table to provide a summary of “Related Work” and add relevant citations to each method. Also, the reviewer appreciates open sourcing the code for reproducibility and others to take the work forward. In conclusion, the reviewer finds the proposed method empirically correct and it is very difficult to find major faults in the paper.

Experimental design

The authors have done a great job by considering several networks on the standard CIFAR and ImageNet dataset to evaluate the proposed pruning method. Also, the authors compared both the metrics, accuracy and FLOPs, with the state-of-the-art works which is important for the pruned models.

Suggestions to Improve: The results section looks good but the authors should also highlight the algorithmic advantage, such as the number of pruning epochs/time, of the proposed method over previous pruning algorithms. This is because there is only a margin of improvement compared to previous methods. The baseline accuracy should also be reported in Tables 1,2,3 and 4 to make it clear for the readers. The pruning methods (methods listed in 3.2.1) against which the authors compare their approach should be summarized in the Related Work section

Questions for clarification:

a) Why did the authors consider random convolution kernel values in Figure 2? It's confusing. It would be good to consider the weights of the pretrained model and use a sample of input images. A clear explanation in rebuttal is required to justify.
b) As the proposed pruning algorithm randomly removes the channels, did authors repeat the experiments with random seeds?
c) What are the advantages and disadvantages of local (EXP-A) and global pruning (EXP-B)? Good to mention this point in the paper.

Validity of the findings

The reviewer has gone through the proposed method in Section 2 and found that it is indeed practical.

Additional comments

Overall, the paper is well written and the results are well reported. The paper requires minor edits to be fully approved.

Cite this review as

Reviewer 2 ·

Basic reporting

The paper presents a new method to prune large neural networks with channel-level random sparsity. Evaluations are performed on CIFAR-10 and ImageNet datasets on models such as VGGNet, and ResNet. The experimental evaluation showed great degree of reduction in FLOPs vs. a minimal accuracy loss.
It has a few typos and hard-to-read text portions. The paper needs rigorous proofreading to fix these.

Experimental design

The experimental design seems interesting. It is, however, not clear how the distribution of sparse channels is selected.

Validity of the findings

The findings seem adequate.

Cite this review as

---

## Round 0.2 · accepted · Accept

I think the authors have done a good job of revising the paper. Yes, it is ready to go.

Reviewer 1 ·

Basic reporting

The authors have significantly improved the paper based on both the reviewer’s comments. They provided justification for all the questions raised by the reviewers and are convincing. Table 1 is a good addition to the paper which provides an overview of related previous methods. The paper is in a good condition for acceptance.

Experimental design

The experiments and related discussion is convincing. The revised manuscript has more points to discuss than earlier.

Validity of the findings

The findings are useful for researchers working on Pruning.

Cite this review as

Reviewer 2 ·

Basic reporting

no comment

Experimental design

no comments

Validity of the findings

no cmments

Additional comments

The revised manuscript addresses the initial comments from reviewers. It is in a much better shape with improved proof reading and clarification of the advantages of the proposed method.

Cite this review as